# Determinants of the Availability of Special Diet Meals in Public Schools from Kraków (Poland): A Cross-Sectional Analysis

**DOI:** 10.3390/nu17243834

**Published:** 2025-12-08

**Authors:** Beata Piórecka, Ewa Błaszczyk-Bębenek, Przemysław Holko, Iwona Kowalska-Bobko, Paweł Kawalec

**Affiliations:** 1Department of Nutrition and Drug Research, Institute of Public Health, Faculty of Health Sciences, Jagiellonian University Medical College, 31-066 Kraków, Poland; ewa.blaszczyk@uj.edu.pl (E.B.-B.); p.holko@uj.edu.pl (P.H.); pawel.kawalec@uj.edu.pl (P.K.); 2Institute of Public Health, Faculty of Health Sciences, Jagiellonian University Medical College, 31-066 Kraków, Poland; iw.kowalska@uj.edu.pl

**Keywords:** special diet, nutrition organization, child nutrition, primary schools, secondary schools, Poland

## Abstract

**Background/Objectives:** Special diets can be required for medical, religious, cultural, or ethical purposes. This study examined the relationship between the organization of school nutrition and the availability of special diet meals among students in public primary and secondary schools in Kraków (Poland). **Methods:** An observational study was conducted in 2022 using a web-based survey targeting managers of primary (*n* = 68) and secondary schools (*n* = 18), as well as parents of attending students (*n* = 1730). Factors associated with providing special diets were analyzed using generalized linear models with robust variance estimators. **Results**: According to school managers, the availability of special diet meals was associated with employing a dietitian responsible for menu planning, the presence of students with disability certificates, students’ participation in school meal programs, and higher per-child nutrition costs. Based on parental reports, 16.01% of all students followed a special diet, most often due to medical recommendations, with a higher prevalence observed among secondary school students (26.7%). Special diets were reported more frequently for children with food intolerances and allergies, obesity, chronic conditions, or disability certificates. Adjusted models also indicated slightly higher probabilities of being on a special diet among students attending secondary schools or sports classes compared with their peers. **Conclusions**: Improving the availability of special diet meals in schools requires legislative action, adequate funding, and institutional support, including investments in kitchen infrastructure and the employment of dietitians. These measures are particularly important in institutions enrolling children with disabilities to ensure equitable access to appropriate nutrition.

## 1. Introduction

Proper nutrition during childhood is essential for supporting growth, development, and long-term health, and schools play a crucial role in shaping children’s dietary habits and ensuring that their individual nutritional needs are met. Some students require special diets due to chronic conditions such as diabetes or food allergies, or as a result of being underweight or overweight. Additionally, dietary modifications may be necessary for religious or cultural reasons or to accommodate vegetarian or vegan preferences. It is assumed that children with various types of disabilities are more likely to require special diets [1]. Depending on specific circumstances, various solutions can be implemented to provide meals for children with special dietary needs. These include modifying meals in the school kitchen for a specific child, ordering catered meals for the child, allowing parents to bring specially prepared meals from home, or designating another public facility by the city or municipality to provide appropriate meals [2]. In Poland, parents are responsible for informing the educational institutions about their child’s health condition and special nutritional needs, typically based on a physician’s certificate. In 2010, it was estimated that approximately 5% of students in Poland had special diets, primarily due to chronic conditions and disorders such as allergies, lactose intolerance, celiac disease, diabetes, cystic fibrosis, and phenylketonuria [3]. Children with food allergies are often unable to consume the same foods as their peers, which may foster feelings of exclusion. They may also perceive parental anxiety during mealtimes and develop fears regarding potential adverse health effects of food intake. Consequently, the quality of life of both children and their parents may be negatively affected [4]. Unlike many Western European countries, Poland currently lacks mandatory regulations requiring schools to provide special diets. This policy gap may lead to inconsistencies in meal provision and contribute to social exclusion or inadequate nutrition among affected students [5]. However, new legal initiatives are being discussed to enhance access to special diets and reduce social disparities related to children’s dietary needs in educational institutions [6].

Alongside the need to accommodate special dietary requirements, the increasing prevalence of overweight and obesity among Polish children represents an additional major public health challenge. According to data from the COSI study coordinated by the WHO, approximately 35% of Polish children aged 7–9 years have excess body weight, including 16% classified as obese. The rates are notably higher among boys, with one in five affected. These findings place Poland among the European countries with the highest rates of childhood obesity [7]. The DINO-PL program (2022–2023) further confirmed the increasing prevalence in this age group, reporting that 27.6% of 7-year-olds and 34.8% of 9-year-olds were affected—again, more commonly boys than girls [8]. Among adolescents, the 2018 HBSC report indicated that 21.7% of Polish youth aged 11–15 years were overweight or obese, up from 19.9% in 2014 [9]. The 2024 HBSC report highlights significant gender differences: among 11-year-olds, 41% of boys and 19% of girls were overweight or obese. By age 15, the prevalence declined to 28% in boys and 13% in girls. Although overall rates decrease with age, boys consistently remain more affected than girls [10].

A 2017 WHO report outlined elements of an integrated program for Poland aimed at intensifying public efforts to reduce childhood obesity, some of which have already been implemented [11]. A school environment that supports healthy eating is essential in countering the aggressive marketing of unhealthy foods. Modifying the nutritional environment in schools can positively influence children’s dietary behaviors [12]. The COSI study further emphasized the urgent need to create healthier food and beverage environments, strengthen health systems to promote healthy diets, and maintain robust child nutrition and obesity surveillance systems [13].

To the best of our knowledge, no current data exist on the availability of special diets for children attending schools in Poland. The aim of this study was to examine the relationship between the organization of school nutrition and the availability of special diets among students in public primary and secondary schools in Kraków (Poland).

## 2. Materials and Methods

### 2.1. Study Design and Data Collection

The study was conducted in November 2022 using an online diagnostic survey administered through a Computer-Assisted Web Interview (CAWI). The target groups included managers of primary and secondary schools, as well as parents/legal guardians of students enrolled in these educational institutions. Data were collected through the Jagiellonian University Medical College survey platform, using distinct questionnaires for school managers and parents. The parent survey was previously evaluated in a pilot study that included a representative subset encompassing 10% of the intended study population. The Department of Social Policy and Health of the Kraków City Office distributed the survey links to school managers. Subsequently, parents received the questionnaire through the schools’ internal mailing systems or via newsletters issued by the school managers. Participation in the study was open to all interested parents of children. An information sheet describing the study, along with a link to the online questionnaire, was included in the message. Complete responses to the questionnaire were obtained from the managers of 68 public primary schools and 18 public secondary schools. In the 2022/2023 school year, there were 114 municipal primary schools with a total enrollment of 48,836 students. During the same academic year, 36 general secondary schools and 34 vocational upper secondary schools collectively educated 33,595 students [14]. Also all parents who completed the questionnaire (regardless of the number of responses provided) were included in the study. The final study group included 96 educational institutions (46.7% of those invited to participate) and a total of 1730 parents or legal guardians of students, representing 4.9% of all children enrolled in the responding schools and 2.1% of children from all invited institutions. A higher response rate was observed among primary schools compared to secondary schools (Figure 1).

The questionnaire for managers addressed the organization of school nutrition and the challenges associated with providing special diets. Managers were also asked about the number and cost of meals served, including those prepared for children with special dietary needs; the structure of kitchen staff and available equipment; and the procedures in place to ensure food and nutrition safety. The questionnaire for parents included items related to the organization of school nutrition and access to special diets. Additionally, it included questions about the child’s health status. Anthropometric data, such as weight and height, were collected. The BMI was calculated and interpreted with national percentile charts [15] and compared to the IOTF cutoffs (the 85th percentile as overweight and 95th and more as obesity) [16].

The component of the study involving parents was approved by the Jagiellonian University Bioethics Committee (No. 1072.6120.198.2022; approval dated 31 August 2022). All procedures involving human participants also received approval from the Ethics Review Board in Humanities and Social and Behavioural Sciences at the University of Helsinki on 24 February 2015 (Statement 6/2015).

### 2.2. Statistical Analysis

All study results were summarized as means with standard deviations (SDs) and medians with interquartile ranges (IQRs) for continuous variables, and as frequencies for categorical variables. Both mean (SD) and median (IQR) were included to provide a transparent description of potentially skewed data distributions. For group comparisons (primary vs. secondary schools), Pearson’s χ^2^ test was applied to categorical variables, and the Wilcoxon rank-sum test was used for continuous variables. However, because of the small number of secondary schoolchildren, substantial clustering by school (1594 primary vs. 136 secondary schoolchildren), and notable differential nonresponse, formal hypothesis testing for unadjusted between-group differences was not pursued.

Logistic regression models with robust variance estimators were used to examine the probability of a student having a special diet and the likelihood of school managers providing such diets. With the exception of the school level, variables were selected using a backward elimination approach with a *p*-value cutoff of 0.2. Only variables with response rates exceeding 70% were included. Interaction terms were incorporated when they meaningfully improved model performance, defined as more than a 10% increase in log-likelihood or pseudo-R^2^. Model selection and assessment were guided by the Box–Cox test, the modified Park test, and log-likelihood statistics. Predictive margins were reported as adjusted means. Average marginal effects (for continuous variables) and contrasts of predictive margins (for categorical variables) were presented as adjusted differences, with standard errors estimated using the delta method.

All respondents who completed the questionnaire were included in the analytical dataset, regardless of the total number of items answered. Missing data were excluded on an outcome-by-outcome basis. No corrections for multiple testing were applied, and statistical significance was set at *p* < 0.05.

Data preparation and statistical analyses were conducted using StataSE 18 (StataCorp, College Station, TX, USA) and OriginPro 2025 (OriginLab Corporation, Northampton, MA, USA). The study followed the guidelines of the Strengthening the Reporting of Observational Studies in Epidemiology (STROBE) Statement [17].

## 3. Results

### 3.1. Organization of Nutrition and Availability of Special Diet Meals from the Perspective of School Managers

The study included large schools, with an average enrollment exceeding 400 students. Secondary schools participating in the study were significantly more likely to have their own kitchens compared to primary schools (55.6% vs. 35.4%, *p* = 0.009). In contrast, primary schools more frequently relied on collective catering provided by an external company, which either rented the kitchen or delivered meals from outside (Table 1). Almost all primary schools (95.6%) reported having menus that included a detailed list of allergenic ingredients or those causing food intolerances, whereas such information was significantly less often included in the menus of secondary schools (72.2%). All primary schools provided menus for both students and their parents, in contrast to secondary schools (66.7%, *p* < 0.001). Nearly all primary schools reported that their nutrition staff assessed menu compliance with current nutritional standards (energy and nutrient intake for the relevant age group), which was again significantly less common in secondary schools (98.5% vs. 72.2%, *p* < 0.001). Dietary meals for students with special nutritional needs were more frequently provided in primary than in secondary schools (54.4% vs. 27.8%, *p* = 0.044) (Table 1).

The provision of dietary meals in schools was significantly associated with the employment of a dietitian for the preparation of the menu, the number of students with a disability certificate (*p* = 0.004), the number of students participating in collective nutrition programs (*p* = 0.010), and the daily cost of nutrition per student (*p* = 0.017; Table 2). Primary schools providing meals for students with special dietary needs more frequently reported having a convection–steam oven among their kitchen equipment (*p* < 0.05). According to school managers, the main reasons why kitchens could not provide meals for students with special nutritional needs were as follows (N = 12): lack of additional staff, including a dietician (*n* =  8, 66.7%); insufficient space or equipment (*n* =  7, 58.3%); and lack of children requiring special meals (*n* = 3, 25%).

### 3.2. Organization of Nutrition and Availability of Special Diet Meals from the Perspective of Parents

The questionnaire was completed by 1594 parents of primary school students and 136 parents of those attending secondary schools. The general sociodemographic characteristics and health status of the study group are presented in Table 3. Most parents of primary school students (92.8%) reported Kraków as their place of residence, whereas 35.3% of parents of secondary school students lived in rural areas. The proportion of parents with a higher education degree was significantly greater among those of primary compared with secondary school students (mothers: 79% vs. 58%; fathers: 63% vs. 31%). Most respondents (67%) described their financial situation as average. Overall, 3.7% of primary school students were classified as obese and 9.2% as overweight. In secondary schools, being overweight affected 15%, and obesity affected 8% of students. The frequency of special diet use as reported by parents is presented in Table 4. Among all primary school students, 15.1% followed a special diet (8.5% due to medical recommendations and 6.6% by parental choice). Food intolerance (4.0%) and food allergy (3.1%) were the most common reasons for following a special diet. The prevalence of special diets was higher among secondary school students (26.7%).

The likelihood of having a special diet was higher among students with food intolerance or allergy compared with those without such conditions, and also among obese students compared with those in other BMI categories (*p* < 0.001). In the total school students, having a special diet was significantly associated with the use of oral medication for a chronic disease (*p* < 0.001) and with possession of a disability certificate (*p* < 0.001) (Table 5). The probabilities of students following a special diet in relation to selected health and behavioral factors are also presented in Figure 2. The probability of having a special diet was slightly higher among secondary school students (0.231 [95% CI: 0.155–0.306] vs. 0.153 [95% CI: 0.137–0.168] in primary school students) and among students attending sports classes (0.204 [95% CI: 0.156–0.251] vs. 0.153 [95% CI: 0.138–0.169] in other classes types). The probability of having a special diet was lower among students who used collective nutrition in a school than among those who did not (0.137 [95% CI: 0.118 to 0.156] vs. 0.195 [95% CI: 0.169 to 0.221]). The probability of adhering to a special diet decreased with increasing snacking frequency between meals, being particularly lower among students who consumed sweetened milk drinks and sweets, but higher among those who snacked on nuts. The probability of adhering to a special diet was also significantly higher among students who bought mineral water from the school shop or a nearby store compared to those who did not.

## 4. Discussion

The study targeted school managers and parents of children attending these educational institutions. According to parents’ responses, 16.1% of students attending primary and secondary schools in Kraków followed a special diet. These findings are consistent with global trends, where dietary restrictions among school-aged children are becoming increasingly common due to rising food allergies and lifestyle-related dietary choices [18]. However, it was found that only half of the primary schools and one-third of the secondary schools offered such meals, indicating that not all students who require a special diet receive it at school. This discrepancy may be explained by the fact that some parents do not report their child’s dietary needs to the school because they lack a medical certificate from a physician. Since 2022, only primary schools in Poland have been required to provide every child with one hot meal per day and to ensure appropriate conditions for its consumption during the school day, such as the availability of canteens. Participation in school meals is voluntary and subject to a fee [19]. In Sweden, as in Poland, parents are responsible for informing educational institutions about their child’s need for a special diet. In this cross-sectional study by Servin et al. [20] found that 19% of preschool children followed special diets, and nearly half of them (47%) did not have a medical certificate. The authors concluded that requiring medical certificates for health-related dietary prescriptions could help reduce unnecessary dietary restrictions among children. In our study, 7.11% of all students required a special diet based on medical recommendations, while 8.9% followed one based on parental choice. Adherence to a special diet was significantly associated with the use of oral medication for chronic conditions and with having a disability certificate. In a cross-sectional study conducted in Croatia, North Macedonia, and Serbia, 14% of children with developmental disabilities followed a special diet, most commonly a gluten-free diet [21].

In the present study, school managers reported that the provision of special diet meals in public schools was associated with the employment of a dietitian for the preparation of the menu, the presence of children with disability certificates, participation in collective nutrition programs, and the daily cost of nutrition per student. A previous study [22] showed that special diet meals were available in 95.2% of nurseries and 60.5% of kindergartens in Kraków, with availability linked to facility type, number of children, the presence of a dietitian, and meal costs. Parents reported that 16.1% of nursery children and 12.7% of kindergarten children received special diets (8.5% medically indicated; 4.2% by parental choice). Similar to the previously mentioned study, among school students, special diets were most often required for medical reasons, such as food allergies and intolerances. As in all EU countries, school menus in Poland are required to provide clear information on substances or products that may cause allergies or intolerances [23]. In line with these requirements, almost all primary schools provided menus listing allergenic or intolerance-inducing ingredients, whereas secondary schools did so far less frequently.

In a study involving 289 parents of children with food allergies in the United States, more than one-quarter reported feeling uncertain or unsafe about their child’s school environment. Although most parents considered existing school policies helpful, they emphasized the need for additional measures, including improved allergen labeling and direct food allergy education for students [24]. The study by Kostecka et al. [25] demonstrated that parents of Polish preschool children were often unprepared to manage their child’s allergy. Parents who were allergic themselves, as well as those of younger children with a broader range of allergy symptoms, achieved higher knowledge scores. The study also confirmed that allergies to multiple food items were associated with the need for long-term elimination diets and difficulties in preparing safe meals, grocery shopping, and maintaining dietary adherence when eating out. The economic and practical challenges associated with preparing dietary meals underscore that food allergies extend beyond nutritional implications, contributing significantly to the socioeconomic strain experienced by households. Kalmpourtzidou et al. [26] highlighted that children with food allergies may be at increased risk of reduced diet quality due to the avoidance of specific foods or food groups. However, an even greater public health concern is the generally low diet quality observed across the broader pediatric population. Findings from the review by Di Profio et al. indicate that diets reducing short-chain fatty acid-producing bacteria, such as gluten-free diet, phenylketonuria, or ketogenic diet, or those characterized by low intake of plant-based foods, may exert negative effects on the host [27].

Caregivers should be supported in promoting dietary diversity, particularly during complementary feeding, and allergy-focused dietary consultations should also address not only allergen avoidance but also the long-term effects of diet on health and disease prevention [28]. Evidence from other countries suggests that structured policies can effectively support dietary needs in schools. Smith et al. [29] found that the introduction of standards under the Healthy Hunger-Free Kids Act of 2010 was associated with improvements in children’s diets and a reduction in dietary disparities across different socioeconomic and ethnic groups. The replacement of home-packed meals with school-provided meals contributed to a greater equalization of dietary quality between children from diverse income and cultural backgrounds. Limited access to special diets may therefore lead to social exclusion of children requiring specific nutrition, as also reported in other studies [29,30]. Additionally, in schools where meals were more diverse and aligned with nutritional guidelines, students reported higher satisfaction with the food offered [31]. Moreover, environmental factors such as food availability and pricing strongly influence children’s dietary choices [32].

Our study identified a correlation between obesity and the likelihood of following a special diet, emphasizing the need for effective interventions in nutrition education and school menu planning. This finding also raises important questions about whether nutrition interventions are appropriately designed to support weight management or whether they inadvertently contribute to unhealthy eating patterns [33]. Day et al. nvestigated primary school staff perceptions of factors influencing the successful implementation and sustainability of healthy lifestyle interventions aimed at addressing childhood obesity. Thematic analysis identified several challenges to effective implementation, including limited time, inadequate training and support, insufficient resources, and staff perceptions of their competency in delivering program activities. Short-term funding and lack of support were noted as barriers to sustainability, whereas strong leadership and the integration of programs into school policies were deemed essential for long-term success [34].

In this study, schools that offered special diets were more likely to employ dietitians for the preparation of the menu and serve a higher number of students with disabilities. However, financial constraints and inadequate kitchen infrastructure were significant barriers, particularly in secondary schools. Research suggests that the cost of special diets, particularly gluten-free options, can be up to 30% higher than that of standard meals, posing affordability challenges [35]. Cost analyses indicate that preparing gluten-free meals incurs higher expenses, a finding supported by other studies [35,36]. In Kraków schools, additional fees for special meals depend on the cost of basic ingredients, potentially leading to inequalities in access to appropriate nutrition. Studies from Spain suggest that sustainable and cost-effective meal planning in school canteens can address this challenge [37]. Moreover, research highlights that school meal subsidies can significantly reduce dietary disparities [29]. A study that assessed the relationship between food expenditure and the nutritional quality of meals was conducted in home-based childcare centers participating in the CACFP. The analysis revealed a significant association between higher daily food expenditures and better nutritional quality of meals, including increased servings of whole grains, fruits, and vegetables [38]. Parents of secondary school students from Kraków reported lower satisfaction with school meal provision compared with parents of primary school students. This may reflect reduced oversight and institutional engagement in secondary education meal programs and accommodation of dietary needs [39]. School policies emphasizing fresh, minimally processed foods contribute to healthier dietary habits [40]. Evidence from Poland and Spain suggests that cross-cultural interventions can improve school meal quality and promote healthier eating behaviors among students [41]. Moreover, improved access to balanced meals can help prevent obesity, particularly among children with disabilities [42].

The results of this study also highlight major challenges in organizing nutrition for children in primary and secondary schools. In particular, limited availability of special diets, along with organizational and financial barriers, affects the feasibility of their implementation. The findings are consistent with previous studies on school meal organization systems in Poland [43] and with international research on the impact of school environments on students’ dietary habits [41,44]. According to the latest report by the Supreme Audit Office (NIK) in Poland, children in educational institutions do not receive meals of adequate quality, mainly due to low food service budgets and a lack of qualified staff. Not all facilities properly organized the preparation and distribution of hot meals—only 21 out of 25 inspected schools and preschools provided them. In 18 of these, special dietary needs, primarily related to food intolerances, were taken into account. Only three institutions employed a dietitian to plan menus, two consulted one occasionally, while in the remaining facilities menus were prepared mainly by administrative or kitchen staff [45]. Educational institutions in Poland should effectively identify and accommodate children’s diverse dietary needs to ensure that no child feels excluded from participation in preschool or school meals. Proposed amendments to national regulations on collective nutrition state that, once implemented, every child requiring an elimination diet for health reasons will be guaranteed a meal provided by the school, prepared in accordance with medical recommendations and food safety standards [6].

### Strengths and Limitations of the Study

To our knowledge, this is the first study to examine the provision of special diets to school-aged children in Poland. The large sample size, including 68 primary schools, 18 secondary schools, and 1730 parents, enabled detailed subgroup analyses. A major strength of the study is the inclusion of two complementary perspectives: that of school managers, who reported on institutional practices, and that of parents, who provided information on children’s dietary needs and health status. However, several limitations should be acknowledged. The small sample size of secondary schoolchildren is a limitation of the study. The cross-sectional design precludes causal inference, and some of the associations observed may be subject to reverse causality.

This was a correlation study; therefore, causal relationships between the variables could not be established. For example, the probability of following a special diet was higher among children with obesity compared with those of normal body weight. However, this does not imply that adherence to a special diet causes obesity; rather, being overweight or obese may lead to the adoption of a special diet. Compared with the general population of Poland, obese children were slightly underrepresented in our sample (obesity, 3.9%; Agresti-Coull 95% CI, 3.1% to 5.0% vs. 5%; overweight, 9.4%, Agresti-Coull 95% CI, 8.1% to 10.9% vs. 10%; overweight and obese, 13.3%, Agresti-Coull 95% CI, 11.8% to 15.0% vs. 15%), while underweight children were overrepresented (17.2%, Agresti-Coull 95% CI, 15.5% to 19.1% vs. 10%) [46].

One potential limitation is the reliance on self-reported data (for example, body weight and height were reported by parents rather than measured), which may introduce reporting bias or lead to missing information. Another limitation relates to the online format of the survey, which may reduce data validity compared with face-to-face interviews and increase the risk of recall bias. The use of an Internet-based questionnaire may also contribute to sampling bias, as individuals with higher education are more likely to participate in online surveys and tend to show greater interest in health-related topics [47]. This is reflected in our sample, in which over 80% of participants had a higher education degree—an overrepresentation compared with the general population. As in many Internet-based studies, the response rate may have been affected by time constraints or by the ease of ignoring an online invitation. Finally, because the research was conducted in Krakow, the findings may not be fully generalizable to other regions of Poland.

## 5. Conclusions

The study underscores the urgent need to improve access to special diets in Kraków public schools. While some institutions have made progress, significant challenges remain in policy, funding, and implementation. Addressing these gaps through legislative action, financial support, and institutional improvements will be crucial in ensuring that all children receive adequate nutrition regardless of dietary restrictions. Particular attention should be paid to the organization of collective nutrition services, including in-house kitchens, in schools accommodating children with disabilities. This would ensure that such schools are able to provide meals tailored to the specific dietary needs of their students. It should be feasible to prepare dietary meals within school, especially in special schools or those with integrated classes. Future research should investigate the long-term health effects of special diets on children’s growth, cognitive development, and overall well-being.

Based on the study findings, it is recommended that legal regulations mandating schools to provide special diets be introduced. These regulations should be accompanied by improvements in kitchen infrastructure and the employment of dietitians in educational institutions. To enhance access for children requiring special diets, subsidized meal programs should also be promoted. Continued research is needed to evaluate the impact of nutrition policies on student health and to assess the effectiveness of implementing healthy eating guidelines in school environments.

## Figures and Tables

**Figure 1 nutrients-17-03834-f001:**
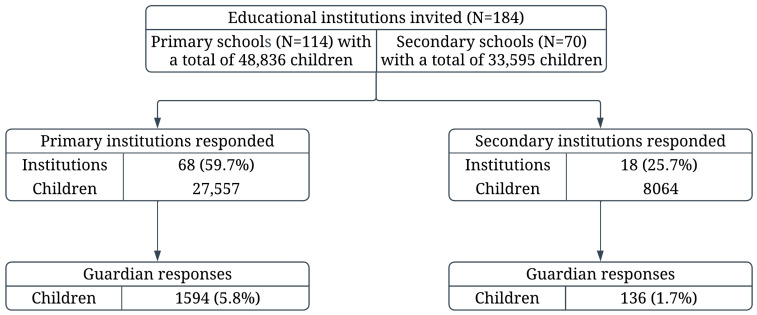
Flow diagram of the study.

**Figure 2 nutrients-17-03834-f002:**
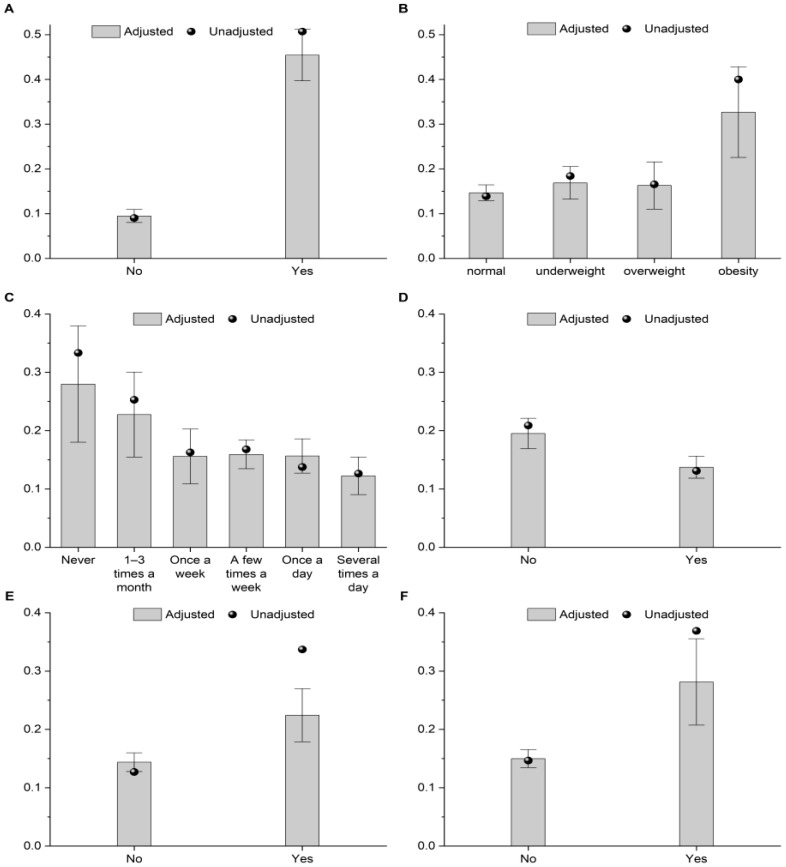
Adjusted (with 95% CI error bars) and unadjusted probabilities of students following a special diet by: food intolerance or allergy (**A**); BMI category (**B**); frequency of eating between meals (**C**); participation in school meals (**D**); use of medications for a chronic disease (**E**); and possession of a disability certificate (**F**).

**Table 1 nutrients-17-03834-t001:** General characteristics of schools participating in the study.

Characteristics		Primary Schools (N = 68)	Secondary Schools (N = 18)	*p*-Value
Number of students enrolled in the schools	Median (IQR), N	396.0 (261.5–520.0), 68	467.5 (137.0–726.0), 18	0.648
	Mean (SD), N	405.3 (214.5), 68	448.0 (323.5), 18	
Number of students with disabilities in the schools	Median (IQR), N	4.0 (1.0–10.0), 67	2.5 (1.0–5.0), 18	0.256
	Mean (SD), N	16.6 (35.4), 67	19.8 (52.2), 18	
Type of meal provision in the schools	own kitchen: *n* (%), N	12 (17.6), 68	10 (55.6), 18	0.009
	rented kitchen: *n* (%), N	26 (38.2), 68	2 (11.1), 18	
	external catering: *n* (%), N	11 (16.2), 68	2 (11.1), 18	
	other: *n* (%), N	4 (5.9), 68	4 (22.2), 18	
Number of students eating meals in the schools	Median (IQR), N	152.5 (115.0–250.0), 68	96.0 (30.0–132.0), 13	0.001
	Mean (SD), N	184.3 (105.4), 68	86.6 (67.6), 13	
Number of students receiving full or partial reimbursement for collective meals	Median (IQR), N	8.0 (3.0–16.0), 68	0.5 (0.0–2.0), 18	<0.001
	Mean (SD), N	17.0 (39.6), 68	1.7 (2.8), 18	
Provision of special diet meals in the schools	*n* (%), N	37 (54.4), 68	5 (27.8), 18	0.044
Number of students receiving special diet meals in the schools	Median (IQR), N	3.5 (2.0–7.5), 36	1.0 (1.0–3.0), 5	0.236
	Mean (SD), N	5.5 (6.6), 36	3.2 (3.9), 5	
Daily cost of nutrition for a student on a special diet (PLN) *	Median (IQR), N	13.0 (11.0–14.0), 29	10.0 (9.0–14.0), 5	0.432
	Mean (SD), N	12.4 (2.7), 29	11.0 (3.4), 5	
Daily cost of nutrition for a student (PLN) *	Median (IQR), N	13.0 (10.0–14.0), 65	12.8 (9.0–15.0), 14	0.801
	Mean (SD), N	12.1 (3.2), 65	12.1 (3.4), 14	
Dietitian employed for the preparation of the menu	*n* (%), N	22 (32.4), 68	4 (22.2), 18	0.405
Menu verified for compliance with current nutritional standards (energy and nutrient requirements for the relevant age group)	*n* (%), N	67 (98.5), 68	13 (72.2), 18	<0.001
Assessment of students’ satisfaction with meals	*n* (%), N	42 (61.8), 68	8 (44.4), 18	0.185
Menu access provided for students and parents	*n* (%), N	68 (100.0), 68	12 (66.7), 18	<0.001
Canteen or dining area available for students	*n* (%), N	66 (97.1), 68	12 (66.7), 18	<0.001

* PLN1 (Polish Zloty) = €0.21 (as of September 2022). IQR, interquartile range; N, number of responses; SD, standard deviation; and N = respondents with non-missing data for the variable/stratum. For categorical variables, *n* = count with the characteristic; % = *n*/N. For continuous variables, summaries are computed on N non-missing observations (no imputation).

**Table 2 nutrients-17-03834-t002:** Association between the probability of providing special diet meals and selected school characteristics.

Characteristic		OR (95% CI)	*p*-Value
Facility type	Primary schools	Reference	
	Secondary schools	0.51 (0.0772 to 3.31)	0.476
Number of students with disabilities in the schools	per 1 additional child	1.03 (1.01 to 1.05)	0.010
Number of students eating meals in the schools	per 1 additional child	1.01 (1.0003 to 1.01)	0.039
Daily cost of nutrition for a student	per 1 PLN increase	1.40 (1.06 to 1.84)	0.017
Dietitian employed	No	Reference	
	Yes	7.76 (1.95 to 30.98)	0.004
On-site kitchen available in the schools	No	Reference	
	Yes	0.48 (0.14 to 1.60)	0.234

Logistic regression model: N = 77; log pseudolikelihood = −39.40; intercept (baseline odds) = 0.007 (95% CI: 0.0001 to 0.416). ORs are adjusted odds ratios. Reference = baseline level for each categorical predictor; ORs for other levels are relative to this level. Overall effect = Wald χ^2^ test for the predictor as a whole; ORs for continuous predictors represent the change in odds per the stated increment.

**Table 3 nutrients-17-03834-t003:** General sociodemographic characteristics and nutritional status of the student population.

Characteristic			Primary School Students (N = 1594)	Secondary School Students (N = 136)
Female sex; *n* (%), N			787 (49.4), 1594	56 (41.2), 136
Age, years	mean (SD), N		10.3 (2.3), 1565	15.6 (2.8), 136
	median (IQR), N		10.1 (8.3 to 12.1), 1565	15.6 (14.6 to 17.2), 132
BMI category; *n* (%), N		Overweight	142 (9.2), 1544	15 (11.5), 131
		Obesity	57 (3.7), 1544	8 (6.1), 131
		Normal weight	1 070 (69.3), 1544	95 (72.5), 131
		Underweight	275 (17.8), 1544	13 (9.9), 131
Students’ physical activity (parent assessment); *n* (%), N		Low	244 (15.3), 1591	49 (36.3), 135
		Moderate	804 (50.5), 1591	58 (43.0), 135
		High	543 (34.1), 1591	28 (20.7), 135
Place of residence; *n* (%), N		Village	115 (7.2), 1594	48 (35.3), 136
		City	1 479 (92.8), 1594	88 (64.7), 136
Number of persons in the household	mean (SD), N		3.9 (0.9), 1587	4.1 (1.1), 135
	median (IQR), N		4.0 (3.0 to 4.0), 1587	4.0 (3.0 to 5.0), 135
Number of underage persons in the household	mean (SD), N		1.7 (0.9), 1582	1.5 (1.1), 135
	median (IQR), N		2.0 (1.0 to 2.0), 1582	1.0 (1.0 to 2.0), 134
Parent with occupational activity; *n* (%), N		Father	229 (14.4), 1587	18 (13.4), 134
		Mother	113 (7.1), 1587	15 (11.2), 134
		Both parents	1 245 (78.4), 1587	101 (75.4), 134
Self-assessed financial situation; *n* (%), N		Average	1 056 (66.4), 1591	95 (70.4), 135
		Above average	417 (26.2), 1591	25 (18.5), 135
		Below average	118 (7.4), 1591	15 (11.1), 135
Education level (mother); *n* (%), N		Basic	8 (0.5), 1590	1 (0.7), 135
		Basic vocational	43 (2.7), 1590	11 (8.2), 135
		Secondary	291 (18.3), 1590	45 (33.3), 135
		Higher	1 248 (78.5), 1590	78 (57.8), 135
Education level (father); *n* (%), N		Basic	32 (2.0), 1589	4 (3.0), 135
		Basic vocational	166 (10.4), 1589	28 (20.7), 135
		Secondary	397 (25.0), 1589	53 (39.3), 135
		Higher	994 (62.6), 1589	50 (37.0), 135
Students using oral medication for a chronic disease; *n* (%), N			247 (15.5), 1592	23 (17.0), 135
Students with a disability certificate; *n* (%), N			87 (5.5), 1591	16 (11.9), 135
Students use collective nutrition at school; *n* (%), N			1034 (64.9), 1594	50 (36.8), 136

BMI, body mass index; N, number of responses; SD, standard deviation; N = respondents with non-missing data for the variable/stratum. For categorical variables, *n* = count with the characteristic; % = *n*/N. For continuous variables, summaries are computed on N non-missing observations (no imputation)

**Table 4 nutrients-17-03834-t004:** Frequency of special diets among primary and secondary school students, as reported by parents (*n,* %).

Special Diet	All (N = 1730)			Primary Schools (N = 1594)		Secondary Schools (N = 136)	
	Overall (*n* = 277)	Parents’ Choice (*n* = 123)	Physician Recommendation (*n* = 154)	Parents’ Choice (*n* = 106)	Physician Recommendation (*n* = 136)	Parents’ Choice (*n* = 17)	Physician Recommendation (*n* = 18)
Any	277 (16.01)	123 (7.11)	154 (8.9)	106 (6.65)	136 (8.53)	17 (12.5)	18 (13.24)
Vegetarian diet	33 (1.91)	33 (1.91)	0 (0)	28 (1.76)	0 (0)	5 (3.68)	0 (0)
Vegan diet	4 (0.23)	4 (0.23)	0 (0)	2 (0.13)	0 (0)	2 (1.47)	0 (0)
Gluten-free diet	45 (2.6)	4 (0.23)	41 (2.37)	3 (0.19)	36 (2.26)	1 (0.74)	5 (3.68)
Diet due to food allergy	55 (3.18)	11 (0.64)	44 (2.54)	10 (0.63)	40 (2.51)	1 (0.74)	4 (2.94)
Diet due to food intolerance	64 (3.7)	17 (0.98)	47 (2.72)	17 (1.07)	46 (2.89)	0 (0)	1 (0.74)
Religious diet	3 (0.17)	3 (0.17)	0 (0)	3 (0.19)	0 (0)	0 (0)	0 (0)
Weight loss diet	41 (2.37)	22 (1.27)	19 (1.1)	19 (1.19)	19 (1.19)	3 (2.21)	0 (0)
Other diet	78 (4.51)	42 (2.43)	36 (2.08)	34 (2.13)	28 (1.76)	8 (5.88)	8 (5.88)

N = respondents with non-missing data for the variable/stratum. For categorical variables, *n* = count with the characteristic; % = *n*/N. For continuous variables, summaries are computed on N non-missing observations (no imputation).

**Table 5 nutrients-17-03834-t005:** Association between student characteristics and the probability of having a special diet.

Characteristic		OR (95% CI)	*p*-Value
Sex	Male	Reference	
	Female	1.34 (0.97 to 1.85)	0.077
Oral medication for a chronic disease	No	Reference	
	Yes	2.15 (1.41 to 3.26)	<0.001
Disability certificate	No	Reference	
	Yes	3.14 (1.77 to 5.57)	<0.001
Food intolerance or allergy	No	Reference	
	Yes	12.47 (8.62 to 18.04)	<0.001
BMI category	Normal	Reference	
	Underweight	1.26 (0.83 to 1.92)	0.273
	Overweight	1.19 (0.66 to 2.15)	0.561
	Obesity	4.48 (2.19 to 9.13)	<0.001
	Overall effect	*-*	0.001
Type of educational institution	Primary school	Reference	
	Secondary school	2.09 (1.07 to 4.07)	0.030
Students attending sports classes	No	Reference	
	Yes	1.66 (1.04 to 2.66)	0.035
Students snack between meals	Never	Reference	
	1–3 times a month	0.67 (0.27 to 1.67)	0.389
	Once a week	0.34 (0.14 to 0.81)	0.015
	A few times a week	0.35 (0.16 to 0.77)	0.009
	Once a day	0.34 (0.16 to 0.75)	0.007
	Several times a day	0.23 (0.10 to 0.54)	0.001
	Overall effect	-	0.011
Snack between meals: sweetened milk drinks and desserts	No	Reference	
	Yes	0.63 (0.42 to 0.95)	0.027
Snack between meals: sweets	No	Reference	
	Yes	0.59 (0.42 to 0.83)	0.003
Snack between meals: nuts	No	Reference	
	Yes	1.57 (1.10 to 2.23)	0.012
Students buy chips in the school shop or in a nearby shop	No	Reference	
	Yes	1.78 (1.00 to 3.16)	0.050
Students buy mineral water in the school shop or in a nearby store	No	Reference	
	Yes	1.45 (1.02 to 2.05)	0.037
Students participate in collective nutrition at school	No	Reference	
	Yes	0.55 (0.39 to 0.77)	0.001

Logistic regression model: N = 1671; log pseudolikelihood = −516.82; intercept (baseline odds) = 0.1787 (95% CI: 0.0111 to 2.8768). ORs are adjusted odds ratios. Reference = baseline level for each categorical predictor; ORs for other levels are relative to this level. Overall effect = Wald χ^2^ test for the predictor as a whole.

## Data Availability

Data are contained within the article.

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
