# Peer review of "Determinants of the Availability of Special Diet Meals in Public Schools from Kraków (Poland): A Cross-Sectional Analysis"

_nutrients, 2025, doi:10.3390/nu17243834_

Round 1

Reviewer 1 Report

Comments and Suggestions for Authors

This is an interesting study with adequate novelty. However, some points should be addresed.

  • In the Introduction section, the authors should add a bit more information concerning the special diets used in chronic conditions and diseases.
  • Are there any data concerning central obesity? This could be a useful information for the readers.
  • A Computer-Assisted Web Interview method was used to collect the data. The validity of such data is lower compared to face-to-face interviews, which may lead to recall bias. This should be stated as a limitation of the study.
  • Body weight and height wereonly reported and not measured by responders. This should also be stated as a limitation of the study.
  •  It is not clear the final response rate of the study population. Please, highlight this index.
  • The small sample size of secondary schoolchildren is also a limitation of the study.
  • All the above limitation should be stated in the relevant section.
  • The normality distribution test should be reported in the statistical analysis section.
  • Only median or mean value should be reported in Table a according to the normality distribution of the variables.
  • P-values should be added into the text in lines 179-187.
  • The Discusion section is written in one huge paragraph. The authors should split this paragraph into 3-4 smaller paragraphs.

Author Response

This is an interesting study with adequate novelty. However, some points should be addresed.

We sincerely thank the Reviewer for carefully evaluating our manuscript and for providing valuable comments that have helped us improve its overall quality. Below, we have provided you with detailed responses to each remark.

Comments 1: In the Introduction section, the authors should add a bit more information concerning the special diets used in chronic conditions and diseases.

Answer: The Introduction section has been supplemented with the recommended information regarding the use of special diets in chronic conditions and diseases.

Comments 2:  Are there any data concerning central obesity? This could be a useful information for the readers.

Answer: Unfortunately, we did not collect information on central obesity in this study.

Comments 3:  A Computer-Assisted Web Interview method was used to collect the data. The validity of such data is lower compared to face-to-face interviews, which may lead to recall bias. This should be stated as a limitation of the study.

Comments 4:  Body weight and height were only reported and not measured by responders. This should also be stated as a limitation of the study.

Comments 5:  The small sample size of secondary schoolchildren is also a limitation of the study. All the above limitations should be stated in the relevant section.

Answer: We have incorporated the indicated study limitations into the Strengths and Limitations of the Study section.

Comments 6:  It is not clear the final response rate of the study population. Please, highlight this index.

Answer: The response rate has been added (2.1% of all children from the invited schools).

Comments 7: The normality distribution test should be reported in the statistical analysis section. Only median or mean value should be reported in Table a according to the normality distribution of the variables.

Answer: We appreciate these suggestions. Our primary analyses use multivariable logistic regression and other generalized linear models, which do not require normality assumptions for covariates. We intentionally report both mean (SD) and median (IQR) in Tables 1 and 3 to provide complementary information on central tendency and variability, especially for skewed variables. Formal normality tests can be overly sensitive in large samples and underpowered in small samples (such as in groups with only 18 schools or 136 children). Therefore, we did not base reporting on a binary “normal/non-normal” classification. To clarify this rationale, we have added an explicit explanation in the Statistical Analysis section.

Comments 8: P-values should be added into the text in lines 179-187.

Answer: P-values have been added. Please note that part of the text presents simple frequencies without hypothesis testing; other relevant p-values were already included in Table 2.

Comments 9: The Discussion section is written in one huge paragraph. The authors should split this paragraph into 3-4 smaller paragraphs.

Answer: The Discussion section has been revised and divided into subsections.

Reviewer 2 Report

Comments and Suggestions for Authors

The manuscript nutrients-3979442 aims to "identify the association of selected risk factors influencing the use of special diets among children attending public primary and secondary schools in Kraków". The manuscript is based on 45 references, most of them published in the last 5 years.

The authors mentioned these "selected" risk factors, which were unknown to the reader, both in the Abstract and the Introduction. Data presented as results in the abstract does not support the conclusions from the same section. 

The manuscript's title is too long and cumbersome.

The same appreciation applies to the whole paper, where children, schoolchildren, students, parents, and managers are mixed, and all the data displayed are blurred. 

The introduction should be clearly organized; the background based on literature data should be selected to clearly lead the reader to the fundamental problem underlying the present work. Moreover, for better understanding, the authors are invited to show extensive literature data, not only limited to Poland. They could provide a general overview and then present evidence of the specific aspects of their country. They are invited to formulate hypotheses to motivate the present study, and then present their objective. 

The used terms are unclear and doubtful; for example, "selected risk factors" (lines 15, 86), "diagnostic survey" (line 92), Computer-assisted Web Interview Method (who guarantees the accuracy of the primary school children's responses?), and "facilities" (line 107) included in the "study sample." 

Materials and methods lack suitable references. 

Moreover, it is not clear who participated in the present study (e.g., school, facilities, children, students, parents, guardians, managers, etc.); the inclusion/exclusion criteria are missing. What's the problem with the diagnostic survey? Does the study involve medical professionals? 

Statistical analysis provides many tools for data analysis (lines 133-159). However, the Results did not evidence their use. Results are blurred, the essential data are presented in 2 tables (Table 1 and Table 3), very agglomerated and hard to decipher. 

The authors are invited to mention in the Results section each statistical method used (as described in the Materials and Methods) and the corresponding data, highlighting their importance in the data analysis of the present study.

What do "overall effect" and "reference" mean in Table 5?

Discussions consist of an enormous epic paragraph (2 pages, lines 250-374). The authors are invited to separate it into subsections to maintain the reader's interest and facilitate data interpretation. 

Moreover, for all uncommonly used terms, the authors are encouraged to include them and corresponding explanations in a table at the end of the manuscript. 

Comments on the Quality of English Language

Major revision

Author Response

The manuscript nutrients-3979442 aims to "identify the association of selected risk factors influencing the use of special diets among children attending public primary and secondary schools in Kraków". The manuscript is based on 45 references, most of them published in the last 5 years.

We sincerely thank the Reviewer for the thorough evaluation of our manuscript and for providing insightful comments that have significantly contributed to improving the quality of our work. Below, we provide detailed point-by-point responses to all remarks.

Comments 1: The authors mentioned these "selected" risk factors, which were unknown to the reader, both in the Abstract and the Introduction. Data presented as results in the abstract does not support the conclusions from the same section. 

Answer: The abstract has been revised according to the Reviewer’s recommendations.

Comments 2: The manuscript's title is too long and cumbersome.

Answer: The title of the manuscript has been simplified and revised.

Comments 3: The same appreciation applies to the whole paper, where children, schoolchildren, students, parents, and managers are mixed, and all the data displayed are blurred. 

Answer: The study focused specifically on school managers and parents of students attending these schools. We have standardized the terminology used throughout the manuscript as much as possible.

Comments 4: The introduction should be clearly organized; the background based on literature data should be selected to clearly lead the reader to the fundamental problem underlying the present work. Moreover, for better understanding, the authors are invited to show extensive literature data, not only limited to Poland. They could provide a general overview and then present evidence of the specific aspects of their country. They are invited to formulate hypotheses to motivate the present study, and then present their objective. 

Answer: The Introduction section has been revised accordingly.

Comments 5: The used terms are unclear and doubtful; for example, "selected risk factors" (lines 15, 86), "diagnostic survey" (line 92), Computer-assisted Web Interview Method (who guarantees the accuracy of the primary school children's responses?), and "facilities" (line 107) included in the "study sample." 

Answer: The description in the abstract and throughout the manuscript has been clarified and reorganized.

Comments 6: Materials and methods lack suitable references. 

Answer: The relevant references have been added to the Materials and Methods section.

Comments 7: Moreover, it is not clear who participated in the present study (e.g., school, facilities, children, students, parents, guardians, managers, etc.); the inclusion/exclusion criteria are missing. What's the problem with the diagnostic survey? Does the study involve medical professionals? 

Answer: The description of the study population and inclusion criteria has been clarified.

Comments 8: Statistical analysis provides many tools for data analysis (lines 133-159). However, the Results did not evidence their use. Results are blurred, the essential data are presented in 2 tables (Table 1 and Table 3), very agglomerated and hard to decipher. 

Answer: We respectfully disagree. The Results section already implements and documents the analytic tools described in the Statistical Analysis section and presents them in a structured three-step sequence for each population (school managers and parents/guardians):

  1. Descriptive summaries (Tables 1 and 3),
  2. Multivariable logistic regressions (Tables 2 and 5),
  3. Post-estimation margins (Table 4).
    Each method is applied as prespecified.

Comments 9: The authors are invited to mention in the Results section each statistical method used (as described in the Materials and Methods) and the corresponding data, highlighting their importance in the data analysis of the present study.

Answer: The Results have been presented in separate sections, describing the information on the availability of special diets obtained from school managers and parents of students, with the relevant statistical methods applied accordingly.

Comments 10: What do "overall effect" and "reference" mean in Table 5?

Answer: Thank you for the query. In Tables 2 and 5, adjusted odds ratios (ORs) from logistic regressions are presented. For categorical variables, one category is defined as the reference; ORs for the remaining categories compare those categories to the reference. The overall effect indicates the joint significance test for the predictor as a whole. Clarifications have been added in the Statistical Analysis section and as a footnote in Table 5.

Comments 11: Discussions consist of an enormous epic paragraph (2 pages, lines 250-374). The authors are invited to separate it into subsections to maintain the reader's interest and facilitate data interpretation. 

Answer: The Discussion section has been revised and divided into subsections.

Comments 12: Moreover, for all uncommonly used terms, the authors are encouraged to include them and corresponding explanations in a table at the end of the manuscript. 

 Answer: The descriptions of all abbreviations used in the manuscript have been added.

Round 2

Reviewer 1 Report

Comments and Suggestions for Authors

The authors have significantly revised and improved their manuscript.

Author Response

The authors thank you for the positive evaluation of the manuscript.

Reviewer 2 Report

Comments and Suggestions for Authors

The reviewer appreciates the authors' efforts to revise the manuscript in accordance with the previous review report. The data provided is clearer. 

Some aspects still need revision:  

1. The second part of the title is doubtful: "Perspectives of Management and Parents" - the authors should remove it or replace it with a more general one, referring to the exploratory character of the current study. 

2. Line 106. What does "proprietary questionnaire" mean? 

3. Please include both questionnaires in the supplementary material. 

4. In summary, the reviewer observed that the term "facility" means "school". The authors are encouraged to replace this term with a more appropriate one: "educational institution." They should check the entire manuscript and make this replacement.

5. Please reformulate the phrase from lines 112-114 for better understanding: "The final study sample included 96 facilities (46.7% of those invited to participate) and a total of 1,730 parents or legal guardians of students, representing 4.9% of all children enrolled in the responding facilities and 2.1% of children from all invited facilities."

6. Human participants (school managers and students' legal guardians) are involved in the study (line 132), and all findings refer to students enrolled in educational programs; thus, "study group" is more suitable than "study sample".

7. Considering the previous observations, the authors should revise Figure 1 to improve data understanding. Please also explain n and N used in Figure 1.

8. Tables 1, 3, 4: Please put the significance of n in the table footer.

9. Table 2. What do "per un additional child" and "per 1 PLN increase" mean?

10. Tables 3 and 5. Please replace "child" with "student" for uniformity and use the plural, as there are numerous, not just one. 

11. Line 214. Please replace Table 3 with Table 4, because the diet details are included in Table 4. Another suggestion is to place Table 3 after its first mention in the manuscript text for better understanding.

12. Table 4. What does "any" mean? Please explain it or replace it with a more explicit term. 

Comments on the Quality of English Language

Moderate revision.

Author Response

Dear Reviewer,

thank you for reviewing our manuscript and for providing valuable recommendations to improve its clarity and overall quality. Below, we present our responses to each of the comments, and we apologize for the insufficient language quality in the original version of the manuscript.

Some aspects still need revision:  

  1. The second part of the title is doubtful: "Perspectives of Management and Parents" - the authors should remove it or replace it with a more general one, referring to the exploratory character of the current study. 

Answer: The title has been revised.

  1. Line 106. What does "proprietary questionnaire" mean? 

Answer: The wording has been revised to ensure that the sentence is clear and understandable.

  1. Please include both questionnaires in the supplementary material. 

Answer: The questionnaires from the system UJCM (in Polish) have been attached.

  1. In summary, the reviewer observed that the term "facility" means "school". The authors are encouraged to replace this term with a more appropriate one: "educational institution." They should check the entire manuscript and make this replacement.

Answer: The manuscript has been reviewed, and the suggested terminology has been implemented.

  1. Please reformulate the phrase from lines 112-114 for better understanding: "The final study sample included 96 facilities (46.7% of those invited to participate) and a total of 1,730 parents or legal guardians of students, representing 4.9% of all children enrolled in the responding facilities and 2.1% of children from all invited facilities."

Answer: The sentence has been revised to: The final study group included 96 educational institutions (46.7% of those invited to participate) and a total of 1,730 parents or legal guardians of students, representing 4.9% of all children enrolled in the responding schools and 2.1% of children from all invited institutions.

  1. Human participants (school managers and students' legal guardians) are involved in the study (line 132), and all findings refer to students enrolled in educational programs; thus, "study group" is more suitable than "study sample".

Answer: The wording has been revised according to the recommendation.

  1. Considering the previous observations, the authors should revise Figure 1 to improve data understanding. Please also explain n and N used in Figure 1.

Answer: In medical sciences, it is generally understood that "N" refers to the total number of patients or schools considered at the upper level, while "n" denotes the number of patients or schools who were actually enrolled, included, selected, or participated in the study. This distinction helps clarify the scope and scale of the data presented in Figure 1 and throughout the manuscript. The figure 1 has been revised in accordance with the recommendation.

  1. Tables 1, 3, 4: Please put the significance of nin the table footer.

Answer: N denotes the number of respondents with non-missing data for the given variable/stratum; n denotes the number of respondents exhibiting the characteristic (category count). Percentages are computed as n/N

  1. Table 2. What do "per un additional child" and "per 1 PLN increase" mean?

Answer: In Table 2, continuous predictors were entered linearly in the logistic model. Reported odds ratios (ORs) reflect the multiplicative change in the odds per stated unit increase of the predictor. Thus, “per additional child” means the OR for each one-unit increase in the number of children, and “per 1 PLN increase” means the OR for each one-złoty increase in cost.

  1. Tables 3 and 5. Please replace "child" with "student" for uniformity and use the plural, as there are numerous, not just one. 

Answer: The table entries have been revised accordingly.

  1. Line 214. Please replace Table 3 with Table 4, because the diet details are included in Table 4. Another suggestion is to place Table 3 after its first mention in the manuscript text for better understanding.

Answer: The wording in the manuscript has been revised according to the recommendation.

  1. Table 4. What does "any" mean? Please explain it or replace it with a more explicit term. 

Answer: We agree that “Any” is ambiguous. In Table 4 we have replaced “Any” with “Overall”